# Population Genetic Study on the European Flounder (*Platichthys flesus*) from the Southern Baltic Sea Using SNPs and Microsatellite Markers

**DOI:** 10.3390/ani13091448

**Published:** 2023-04-24

**Authors:** Marcin Kuciński, Magdalena Jakubowska-Lehrmann, Agnieszka Góra, Zuzanna Mirny, Katarzyna Nadolna-Ałtyn, Joanna Szlinder-Richert, Konrad Ocalewicz

**Affiliations:** 1Department of Marine Biology and Ecology, Institute of Oceanography, University of Gdansk, Piłsudskiego Ave. 46, 81-378 Gdynia, Poland; 2Department of Fisheries Oceanography and Marine Ecology, National Marine Fisheries Research Institute, Kołłątaja 1 Street, 81-332 Gdynia, Poland; 3Department of Food and Environmental Chemistry, National Marine Fisheries Research Institute, Kołłątaja 1 Street, 81-332 Gdynia, Poland; 4Department of Fisheries Resources, National Marine Fisheries Research Institute, Kołłątaja 1 Street, 81-332 Gdynia, Poland

**Keywords:** population genetics, genetic diversity, genetic structure, microsatellite DNA, SNP, *Platichthys flesus*

## Abstract

**Simple Summary:**

The European flounder (*Platichthys flesus*), which is closely related to the recently discovered Baltic flounder (*Platichthys solemdali*), is currently the third most commercially fished species in the Baltic Sea. The aim of this study was to obtain information on the current patterns of genetic diversity and the population structure of the European flounder and to verify whether the Baltic flounder is present in the southern Baltic Sea. Moreover, we aimed to verify whether the observed decline in the body condition indices of the species in the Baltic Sea might be associated with adaptive changes in its gene pool due to increased fishing pressure. The examined European flounder specimens displayed a high level of genetic diversity and represented a single genetic cluster. The applied molecular markers did not detect the presence of the Baltic flounder among the fish sampled from the studied area. Correlation analysis between genetic and morphological characteristics did not detect any signs of directional selection or density-dependent adaptive changes in the gene pool of the examined fish.

**Abstract:**

The European flounder (*Platichthys flesus*), which is closely related to the recently discovered Baltic flounder (*Platichthys solemdali*), is currently the third most commercially fished species in the Baltic Sea. According to the available data from the Polish Fisheries Monitoring Center and fishermen’s observations, the body condition indices of the species in the Baltic Sea have declined in recent years. The aim of the present study was to obtain information on the current patterns of genetic variability and the population structure of the European flounder and to verify whether the Baltic flounder is present in the southern Baltic Sea. Moreover, we aimed to verify whether the observed decline in the body condition indices of the species in the Baltic Sea might be associated with adaptive alterations in its gene pool due to increased fishing pressure. For this purpose, 190 fish were collected from four locations along the central coastline of Poland, i.e., Mechelinki, Władysławowo, the Vistula Lagoon in 2018, and the Słupsk Bank in 2020. The fish were morphologically analyzed and then genetically screened by the application of nineteen microsatellite DNA and two diagnostic SNP markers. The examined European flounder specimens displayed a high level of genetic diversity (*PIC* = 0.832–0.903, *I* = 2.579–2.768). A lack of significant genetic differentiation (*Fst* = 0.004, *p* > 0.05) was observed in all the examined fish, indicating that the European flounder in the sampled area constitutes a single genetic cluster. A significant deficiency in heterozygotes (*Fis* = 0.093, *p* < 0.05) and overall deviations from Hardy–Weinberg expectations (H-WE) were only detected in fish sampled from the Słupsk Bank. The estimated effective population size (*Ne*) among the sampled fish groups varied from 712 (Słupsk Bank) to 10,115 (Władysławowo and Mechelinki). However, the recorded values of the Garza–Williamson indicator (*M* = 0.574–0.600) and the lack of significant (*p* > 0.05) differences in *Heq* > *He* under the *SMM* model did not support the species’ population size changes in the past. The applied SNP markers did not detect the presence of the Baltic flounder among the fish sampled from the studied area. The analysis of an association between biological traits and patterns of genetic diversity did not detect any signs of directional selection or density-dependent adaptive changes in the gene pool of the examined fish that might be caused by increased fishing pressure.

## 1. Introduction

The European flounder (*Platichthys flesus*) belongs to the Pleuronectidae family that is native to the coastline of the northeastern Atlantic Ocean and the Baltic Sea [1]. It is a euryhaline fish species that leads a demersal lifestyle, feeding on benthic invertebrates and small fish [2,3,4]. For a long time, two distinct ecological forms of the European flounder in the Baltic Sea have been considered. These two forms have shown different reproductive behaviors, i.e., pelagic and demersal [5]. Based on recent molecular evidence, the demersal form is now considered a new cryptic species: the Baltic flounder (*Platichthys solemdali*) [6].

The European flounder is an economically important, non-quota species fished in the Baltic Sea [7]. According to data from the International Council for the Exploration of the Sea (ICES), after the recent collapse of the cod (*Gadus morhua*) fishery, the European flounder has become the third most exploited species in the Baltic Sea [8]. Unfortunately, the available information shows about a 50% decrease in the total landings of the species in the Baltic Sea between 2016 and 2021 [8]. Other data show an almost 90% decline in the proportion of the European flounder in the total landings of flounder in the Gulf of Finland between 1980 and 2018 [9]. In addition, concerning reports from fishermen indicate that the body condition indices of the species have been decreasing, and caught fish are becoming smaller and thinner.

To date, at least three main hypotheses have been proposed to explain the observed decline in the population size/abundance and body condition indices of flounder in the Baltic Sea. The first hypothesis proposes that this decline is caused by the increasing deterioration of environmental conditions due to climate change and eutrophication, which shrink the species’ spawning grounds; cyanobacterial blooms in the nursery grounds; and higher rates of disease occurrence in the fish caused by invasive parasitic organisms [5,9]. The second hypothesis proposes that the decline is caused by growing competition for food and space from other native species that are better adapted to inhabit lower salinity environments, such as cyprinids and Baltic flounder [5,9]. The third hypothesis proposes that the decline is caused by adaptive genetic changes in the species’ gene pool toward limited growth rate and body size, and/or reduced population’s body condition indices due to fishery-mediated heavy selective pressure [10,11,12].

A research project recently launched by the National Marine Fisheries Research Institute (Poland) has indicated that the observed decrease in the flounders’ body condition indices in the Baltic Sea may be associated with environmental changes caused by climate change and anthropogenic pressure [13]. However, the research project did not include genetic examination of the species. This is significant because available data on the patterns of genetic diversity and the population structure of the European flounder were published more than a decade ago. Consequently, there is a lack of comprehensive molecular research examining the composition of the flounder species in the southern Baltic Sea. Moreover, the molecular research of Momigliano et al. [6,9] assessing the stock composition of flounders did not detect any signs of Baltic flounder presence in the southern Baltic Sea. In addition, the research of Momigliano et al. mainly focused on fish from the northern Baltic Sea and did not include areas located close to the Vistula River mouth, which potentially provides suitable environmental conditions for the Baltic flounder [6,9]. Because the research of Svedäng and Hornborg [12] demonstrated the strong impact of selective fishing on density-dependent adaptation in the body condition indices of Atlantic cod in the Baltic Sea, we might expect a similar effect to occur in the European flounder in recent years. If this selective effect was strong enough, it may be detected through the analysis of population genetic and morphological data. Moreover, in a severely overexploited population in which the number of individuals is drastically reduced, there is a higher likelihood of harmful alleles becoming fixed in the population. Harmful alleles are genetic variants that decrease an organism’s fitness; their fixation in a population can lower its viability, increasing the risk of extinction [5].

Therefore, the purpose of the present research was to obtain information on the current genetic status of the European flounder from the southern Baltic Sea and compare it with previous reports [14,15]. The study also sought to verify whether there are any signs of Baltic flounder presence in the southern Baltic Sea that could indicate species composition changes in relation to previously reported data [9]. Moreover, the study aimed to verify whether the observed decline in the species’ body condition indices might be associated with adaptive responses in its gene pool associated with the increased impact of fishing pressure in the Baltic Sea.

## 2. Materials and Methods

### 2.1. Fish Sampling and Morphological Data Collection

In total, 190 fish were sampled from four sampling sites along the central coastline of Poland, i.e., Mechelinki (54°37′07.0″ N 18°33′01.4″ E) (N = 50), Władysławowo (54°48′17.6″ N 18°24′33.1″ E) (N = 50), the Słupsk Bank (55°02′22.6″ N 16°22′03.0″ E) (N = 50), and the Vistula Lagoon (54°21′00.0″ N 19°33′00.0″ E) (N = 40) (Figure 1). Samples from Mechelinki, Władysławowo, and the Vistula Lagoon were collected in 2018 with the cooperation of local fishermen using square mesh panels. Samples from the Słupsk Bank were gathered in 2020 using the research vessel, *Oceanograf*, (University of Gdańsk). The fish sampled from Mechelinki, Władysławowo, and the Słupsk Bank represented mature fish that were at least 3+ years old. The fish sampled from the Vistula Lagoon were all young and immature, aged 1+ or 2+ years, indicating that the latter sampling site may be the nursery ground of the species.

The collected fish were measured to determine total body length (±0.1 cm) and mass (±0.01 g). The fish were then dissected to determine their sex and liver mass (± 0.01 g) (Appendix A). For the molecular analysis, fin clips were collected from each fish and stored in individual Eppendorf tubes filled with 96% ethanol.

### 2.2. DNA Isolation, Microsatellite DNA Markers Amplification and Genotyping

Genomic DNA were extracted and purified using the DNeasy Blood & Tissue Kit (Qiagen, Hilden, Germany) following the manufacturer’s protocol. The quality of the isolated DNA was checked using electrophoresis in 1.5% agarose gel.

Nineteen microsatellites that displayed satisfactory amplification performance were used for the genetic population study on the European flounder. Eleven of these had been previously used in studies on the species in the Baltic Sea: *StPf1001*, *StPf1002*, *StPf1003*, *StPf1004*, *StPf1005*, *StPf1006*, *StPf1015*, *StPf1016*, *StPf1022*, *PL142*, *PL167* [14,15]. The remaining eight markers, *FLAG5-83*, *FLAG4-71*, *FLAG2-76*, *FLAG4-65*, *FLAG8-37*, *Nplaf-33*, *FLAC4-67*, *FLAC4-69*, originated from other studies on flounder along the Atlantic coast [16,17] (Table 1). Nested PCRs were performed in multiplexes according to the protocol described by Schuelke [18] and Nakano et al. [19].

PCRs were prepared in 12.5 µL volume mixtures that contained 1X GoTaq^®^ Hot Start Green Master Mix (Promega Corporation, San Luis Obispo, CA, USA) and varying primer concentrations, according to the recommendations described by Schuelke [18] (Appendix A). Around 10 ng of template DNA was used for each amplification. Nuclease-free water was added to the reaction mixture to obtain the desired final volume. Amplification was performed using a Thermal Cycler SimpliAMP (Applied Biosystems, Foster City, CA, USA) under the following conditions: 95 °C for 4 min, followed by 34 cycles of 95 °C for 30 s, 59 °C for 90 s, and 72 °C for 90 s, then eight cycles of 95 °C for 30 s, 55 °C for 90 s, and 72 °C for 90 s, and a final elongation step at 65 °C for 30 min. To reduce the probability of genotyping errors and the possible effect of a null allele, each homozygote was re-amplified in simplex reactions.

Genotyping was conducted using the Applied Biosystems 3130 Genetic Analyzer against the GeneScan 600 LIZ size standard (Applied Biosystems, Foster City, CA, USA). Electropherograms were analyzed using Peak Scanner Software ver. 1.0 (Applied Biosystems).

### 2.3. Amplification and Genotyping of SNP Markers

To analyze stock composition, we used two diagnostic SNP markers (SNPs 886_19 and 3599_4) that have been previously identified as being under divergent selection in the Baltic and European flounders [6,9]. The selected SNPs have been demonstrated to be almost fixed in both species, enabling the unambiguous determination of the species [20]. Both the SNP loci were amplified separately using primers and conditions described by Momigliano et al. [6]. To determine each SNP’s variants, the amplified DNA fragments were sequenced in both directions according to Sanger’s method using the Applied Biosystems 3130 Genetic Analyzer. Ten randomly selected fish from each sampling location were genotyped using the applied SNP markers.

### 2.4. Data Analysis

The obtained microsatellite raw data were checked for the presence of microsatellite null alleles, inconsistent values, scoring errors, and large allele dropouts using Micro-Checker software ver. 2.2.3 [21]. The observed microsatellite allele frequency and number (*Ao*), Shannon’s index (I), the polymorphism information content (*PIC* value), and the inbreeding coefficient (*Fis*) were calculated using GenAlEx (version 6.5) and PowerMarker (version 3.25) [22,23]. The Ewens–Watterson–Slatkin exact neutrality test, Beaumont and Nichols’ detection test of loci under selection, the Garza–Williamson index (M-ratio), the observed heterozygosity (*Ho*), and the expected heterozygosity (*He*), as well as the Hardy–Weinberg equilibrium (H–WE) test for each locus were performed using Arlequin software (version 3.5) [24]. To assess the global H–WE, Fisher’s and Smouse’s multilocus analysis methods were employed, using the Genepop (version 2.9.3.2) and Popgene (version 1.3.2) [25,26] computer programs. Significance levels for H–WE and the *Fis* indicator were adjusted using the sequential Bonferroni correction [27].

The genetic structure and differentiation among the fish from each sampling site were assessed using the estimation of genetic differentiation index (*Fst*) and the analysis of molecular variance (AMOVA) using Arlequin software (ver. 3.5). The genetic heterogeneity among tested fish groups was examined using ONCOR (ver. 2.0) and GeneClass (ver. 2.0) software [28,29]. The leave-one-out method and individual assignment tests were applied. Additionally, a set of analyses was carried out by applying algorithms that search for putative genetic clusters (*K*) without a priori information on the origin of the examined fish. For this purpose, we used two computer programs that apply different clustering principles, i.e., Structure (ver. 2.3.4) and Flock (ver. 3.1.) [30,31]. The first program applies a Bayesian clustering analysis method, detecting the most probable number of *K* within the sampled fish group. Ten runs were completed for each tested number of *K* (*K* = 1–8), setting the admixture model with 250,000 burn-in periods and one million Markov chain Monte Carlo (MCMC) replicates. The *ΔK* method of Evanno et al. [32] was used to estimate the most probable number of genetic clusters using Structure Harvester online software [33]. Flock software uses a frequentist and partly deterministic method that relies on iterative re-allocation. *K* is identified using Plateau analyses that are based on the repetition of identical cluster solutions. Additionally, the representation of the genetic structure independent of H–WE optimization––i.e., allele sharing distances (*DAS*), principal component analysis (*PCA*), and principal coordinate analysis (*PCoA*)––was also carried out using Populations (ver. 1.2.32) [34], R package Adegenet (ver. 2.1.5) [35], and GenAlEx software, respectively.

To delineate the historical demography of the studied fish group, effective population size (*Ne*) was estimated using the NeEstimator computer program (ver. 2.01) [36]. The single-sample method based on random linkage disequilibrium was applied. To obtain the best ratio between precision and bias, the criterion for excluding rare alleles of *P_crit_* = 0.02 and 0.05 was chosen, in line with Waples and Do [37]. Additionally, to detection of past population size decline in the studied fish groups was carried out using Bottleneck software (ver. 1.9) [38]. For this purpose, the equilibrium was tested using an infinite alleles model (*IAM*), stepwise mutation model (*SMM*), and two-phase mutation model (*TPM*). This method assumes that the excess of expected heterozygosity (*He*) over expected heterozygosity under a mutation–drift equilibrium (*Heq*) in the population indicates the population size reduction and associated bottlenecks.

The obtained DNA sequences of genomic regions bearing analyzed SNP markers were viewed using BioEdit software (ver. 7.2.5) [39] to recover the respective genotypes for each analyzed fish. The retrieved SNP genotype data were then manually compared with genotype profiles established for European and Baltic flounders by Momigliano et al. [6,14].

An analysis was conducted to identify an association between biological traits (i.e., sex, total body length, body mass, liver mass) and the pattern of genetic diversity (i.e., allele frequency, individual pairwise genetic distance, *PIC*, *I*, *Ho, Fis*) to assess the possible effect of selective pressure on the genetic structure of the studied fish. For this purpose, the recorded meristic parameters of the studied fish were transformed into relative values, i.e., Fulton’s condition factor (*K*) and the hepatosomatic index (*HSI*) [40,41]. Next, the sampled fish were divided into series of separate groups according to sex, the values of Fulton’s condition factor (*K*; (total weight × total lenght^−3^) × 100), and the hepatosomatic index (*HSI*; (liver weight × total weight^−1^) × 100) (Appendix A). All the abovementioned genetic parameters were then recalculated for each group. A non-parametric Kruskal–Wallis post hoc test analysis and a set of multidimensional analyses, such as principal components analysis (*PCA/PCoA*), correspondence analysis (*DA*), and discriminant function analysis (*DFA*), were implemented using Statistica (ver. 12.0) (StatSoft Company, Kraków, Poland) and GenAlEx software. These analyses were conducted to check for any significant (*p* < 0.05) associations in the estimated values of genetic parameters with any of the established fish groups as a sign of possible density-dependent genetic adaptation toward body condition indices. For *PCoA* analysis, individual pairwise genetic matrices for all analyzed fish were calculated, then each fish in the matrix was labeled according to the series group in their division. Lastly, a 2-dimensional plot was constructed. Because fish from the Vistula Lagoon displayed significant differences in terms of cohort composition, the analysis of correspondence between biological traits was carried out for all fish except for those sampled from the Vistula Lagoon.

## 3. Results

The Micro-Checker software did not detect any data consistency failures associated with stutter miscalls, scoring errors, or allelic dropout in the collected microsatellite DNA data. The estimated frequencies of possible null alleles across all the examined loci were below 5%, excluding the influence of possible amplification issues on the estimated population genetic parameters [42,43]. The overall number of alleles at the individual locus ranged from 6 to 47 (average = 23.9) (Appendix A). The Ewens–Watterson–Slatkin test was applied, showing the selective neutrality of the applied microsatellite markers.

The mean number of alleles per locus among fish from each sampling site varied between *Ao* = 17.9 and 19.4. The average values of polymorphism information content (*PIC*) and Shannon’s index ranged from 0.832 to 0.903 and from 2.579 to 2.768, respectively. The multilocus analysis revealed significant (*p* < 0.05) overall values of the fixation index (*Fis* = 0.074–0.093) in fish sampled from Mechelinki and the Słupsk Bank. However, after Bonferroni correction, significant values were observed only in fish from the latter sampling site. The recorded values of the fixation index (0.031 and 0.036) in fish sampled from Władysławowo and the Vistula Lagoon were not significant (*p* > 0.05). The mean values of observed heterozygosity (*Ho*) and expected heterozygosity (*He*) varied from 0.751 to 0.815 and from 0.841 to 0.847, respectively. Significant differences between values of observed heterozygosity (*Ho*) and expected (*He*) heterozygosity were observed across four (Władysławowo and the Vistula Lagoon) and five loci (Mechelinki and the Słupsk Bank). After Bonferroni correction, significant differences were observed only in one locus (Słupsk Bank). The H–WE probability global tests detected significant (*p* < 0.05) deviations from the H–WE expectations in fish sampled from Mechelinki and the Słupsk Bank. These deviations remained significant after Bonferroni correction only for fish from the latter group.

The AMOVA analysis revealed that most of the genetic variability occurred within individuals (93.6%), while only 0.04% occurred between fish groups originating from each sampling site. Similarly, the pairwise *Fst* genetic differentiation between the examined groups of fish from each sampling site varied from 0.001 to 0.003, while the global test of differentiation revealed an insignificant value (*Fst* = 0.004, *p* > 0.05). The individual assignments test showed a low level of fish classification correctness for each sampling location, ranging from 26% (Władysławowo) to 40% (Słupsk Bank) (quality index = 31.79%). The resolved individual’s tree based on *DAS* genetic distances, *PCoA,* and *PCA* analyses revealed that all the examined fish formed one common genetic cluster without any signs of significant genetic differentiation (Figure 2). Similarly, the results for the individual multilocus genotype based on Bayesian analyses and Flock software did not identify any signs of genetic clustering in the examined fish. The results of the diagnostic SNP marker analysis did not detect any signs of Baltic flounder presence among the studied fish (Appendix A).

The estimated effective population size (*Ne*) for all examined fish equaled *Ne* = 9101 (95% CI = 3425-infinite, *P_crit_
*= 0.05). In the fish sampled from Mechelinki, Władysławowo, and the Vistula Lagoon in 2018, the estimated value of *Ne* equaled *Ne* = 10115 (95% *CI* = 2113-infinite, *P_crit_
*= 0.05). In the fish sampled from the Słupsk Bank in 2020, the estimated value of the indicator was considerably lower, i.e., *Ne* = 712 (95% *CI* = 207-infinite, *Pcrit* = 0.05). Significant (*p* > 0.05) *He* > *Heq* differences were recorded only under the infinite alleles model (*IAM*) and the two-phase mutation model (*TPM*), where nine loci (*FLAG2-76*, *FLAG4-65*, *Stpf1002*, *Stpf1006*, *Stpf1015*, *Stpf1016*, *PL142*, *PL167*, and *FLAC4-69*) exhibited significant heterozygosity excess (Appendix A). The two-tail Wilcoxon test detected a significant (*p* < 0.05) overall *He* > *Heq* excess only under the *IAM* mutation model. The mean values of the *M*-ratio in all examined fish groups were similar, ranging from 0.574 to 0.647. The lowest values of this indicator (*M* < 0.5) were recorded for *FLAG5-83*, *FLAG4-65*, *FLAG8-37*, *Nplaf-33*, *Stpf1001*, *Stpf1002*, *Stpf1015*, *Stpf1016*, and *FLAC4-69* (Table 2).

The Beaumont and Nichols’ method for the selection of signatures implemented using Arlequin software [24] did not detect any microsatellite loci, with outlier *Fst* values at the 95% confidence level in the examined fish groups. The performed statistical tests did not detect any significant differences (*p* > 0.05) in estimated genetic parameters among the predefined series of fish groups according to sex, the values of Fulton’s condition factor, and hepatosomatic index body condition indices in all comparisons. The performed multidimensional analyses showed no association between genetic parameters and biological characteristics in the examined fish (Figure 3).

## 4. Discussion

Marine ecosystems are dynamic complexes of biological, environmental, and anthropogenic factors such as life history, demography, inter-generic interactions, hydrology, climate, pollution, and fisheries, all of which shape the genetic diversity and structure of marine fish populations [44,45]. Selective pressures associated with environmental and anthropogenic factors may lead to the loss of genetic diversity in fish. This decreases their adaptation abilities to changing ecological conditions, making them more vulnerable to environmental stressors, disease, and other threats [46,47,48]. Moreover, in severely overexploited populations, the fixation of harmful alleles through genetic drift may decrease population viability, increasing the risk of extinction [44,46]. Thus, spatiotemporal genetic-based monitoring of marine resources is an important tool in sustainable management, providing information on the species/population’s past and current life history, degree of relationship and patterns of genetic diversity, and population structure [5,49,50,51].

The results of our study show that despite a significant increase in fishing pressure on the European flounder in the Baltic Sea over the last 20 years, the recorded values of genetic variability indicators (*Ao* = 17.9–19.4, *PIC* = 0.832–0.903, *I* = 2.579–2.768) in the flounder from all sampling sites were high, and comparable with data obtained over a decade ago from fish sampled in other parts of the Baltic Sea [15,16]. This temporal stability in the levels of estimated genetic variability parameters in the examined species might be caused by the large gene flow between flounder from the Baltic and the North Sea, as previously hypothesized by other researchers [15,16,52]. A similar mechanism of genetic diversity stabilization over time has also been reported in other heavily exploited marine fish species [53,54]. The estimated levels of genetic variability in the fish examined in our study were high and comparable with those reported in many other flounder species [55,56,57,58]. This suggests that the currently observed decline in the body condition indices of the European flounder in the Baltic Sea might not be directly associated with the reduction of the species’ genetic variability but rather, with the qualitative changes in the gene pool caused by directional selection pressure associated with environmental changes and intensive fishery exploitation [12]. However, the applied method for loci selection signatures and the correlation analysis between genetic and biological traits did not reveal signs of directional selection or density-dependent adaptative changes in the gene pool of the European flounder specimens examined in this study. The lack of a significant correlation between the patterns of genetic diversity and the reduced body condition indices of fish was also reported by Poćwierz-Kotus et al. [59] in Atlantic cod from the southern Baltic Sea.

The available studies indicate that the genetic structure of flounder populations along the European coastline from the Bay of Biscay to northern Norway is mainly shaped by the latitudinal temperature gradient; however, factors such as migratory behavior, the strict association of juveniles with estuaries or freshwater habitats, and the adaptive response to environmental pollution may also play an important role in shaping the genetic structure of the species across its distribution range [10,52,60,61,62]. To date, genetic population studies on the European flounder have reported low (*Fst* = 0.005–0.090) but significant and stable levels of genetic differentiation, supporting the existence of at least eight genetic clusters of the species in Europe, i.e., (1) Iberian Peninsula, (2) Bay of Biscay, (3) English Channel, (4) North Sea and Irish Sea, (5) Baltic Sea, (6) Faroe Islands, (7) central coast of Norway, and (8) northern coast of Norway [15,16,52,63,64]. The genetic analyses performed in our study did not detect any signs of significant genetic structure in any the studied fish from the sampled areas. Similar levels of genetic homogeneity (*Fst* = 0.005–0.015) have been reported among European flounder populations in the North Sea and Irish Sea [15,16]. The results of the diagnostic SNP marker analysis did not find any signs of Baltic flounder presence among the studied fish. This may suggest that the observed decline in the body condition indices of the European flounder in the Baltic Sea is not associated with the presence and competition for food and space with the Baltic flounder in the studied area. Nevertheless, the lack of any signs of Baltic flounder in the Vistula Lagoon, which theoretically provides suitable conditions for this species, is intriguing and suggests the presence of an unknown environmental barrier that prevents Baltic flounder from entering the southern Baltic Sea. Existing biological information, together with the age analysis of the fish sampled in this study, suggest that the Vistula Lagoon serves as an important nursery ground for juvenile European flounder [65].

In contrast to freshwater fish species, the estimated effective population size (*Ne*) of marine fish populations is frequently several times lower than census data. This makes them particularly vulnerable to the loss of genetic diversity due to population size bottlenecks caused by the impact of increased fishing pressure or changes in the environment [46,47,48,66,67,68,69]. Thus, estimating and tracking effective population size is a powerful tool in the conservation genetics of marine fish insofar as it can be used to determine fish population/stock susceptibility to the loss of genetic integrity. The estimated effective population size (*Ne*) of fish from Władysławowo, Mechelinki, and the Vistula Lagoon sampled in 2018 was high, and comparable with the previously estimated values of *Ne* for flounder populations in the North and Irish Seas [15]. Moreover, the estimated values of *Ne* for fish sampled from the Słupsk Bank in 2020 displayed visibly lower values of this indicator, i.e., *Ne* = 712. This finding might indicate a change in the population size of European flounder in the southern Baltic Sea. However, this conclusion is very speculative and requires further investigation. Interestingly, a similar number of differences in *Ne* values was reported in overexploited plaice populations from Iceland (*Ne* = c.a. 2000) and landlocked populations of Atlantic cod from Mogilnoe Lake (*Ne* = c.a. 200), where the decrease in this parameter was assumed to be accelerated by the fish non-random mating and the associated cryptic mating structure [48,70].

The uncoupling between observed heterozygosity (*Ho*) and expected heterozygosity (*He*) may indicate the presence of a recent demographic disturbance [68,69,70,71]. Numerous studies demonstrate that significant heterozygosity deficiency in marine fish may be associated with population size changes, inbreeding, the Wahlund effect, homoplasy, or selection [72,73,74]. In this study, the fish group from the Słupsk Bank was not in the Hardy–Weinberg equilibrium, which is similar to results reported a decade ago for European flounder sampled from other locations in the Baltic Sea [15,16,52]. The recorded values of the fixation index were the highest in fish sampled from the Słupsk Bank in 2020 (*Fis* = 0.093, *p* < 0.05), suggesting a recent reduction in population size in the sampled area. However, the recorded values of the Garza–Williamson indicator (*M* = 0.574–0.600) and the lack of significant differences of *Heq* > *He* under the *SMM* model do not support this conclusion [75]. Moreover, null alleles are known to be a frequent cause-factor that may be responsible for observed heterozygosity deficits. However, in the present study, all the algorithms used in the Micro-Checker software to estimate null alleles excluded this eventuality, as the recorded levels of excess homozygotes greatly exceed the estimated frequencies of possible null alleles. The lack of significant genetic differentiation among the examined fish groups seems to exclude the possibility of the Wahlund effect, which is congruent with the conclusions of Hemmer-Hansen et al. [15]. Thus, a more comprehensive study that includes series of retrospective cohort-labeled samples is required to confirm the species population size dynamics observed in the present study and to track the population over time.

## 5. Conclusions

In the present study, the recorded levels of genetic diversity among the studied European flounder from the southern Baltic Sea were high and temporally stable in relation to the data published about a decade ago. Moreover, the recorded levels of homozygosity were similar to previously reported genetic data. The results indicate that the European flounder from the southern Baltic Sea represent a single and homogenous genetic cluster. Significant differences in the effective population sizes (*Ne*) between fish sampled in 2018 and those sampled in 2020 might indicate recent changes in the population size of the European flounder in the southern Baltic Sea. However, this conclusion is speculative and requires further studies to be verified. The results do not support the hypothesis of adaptive genetic changes in the flounder’s gene pool toward a limited growth rate, body size, and/or reduced population body condition indices caused by fishery-mediated heavy selective pressure on the species’ populations in the southern Baltic Sea. Our study did not detect the presence of the Baltic flounder in the southern Baltic Sea. Broader geographic and time scale research based on more sensitive genetic analysis methods, such as whole genome sequencing or gene expression, is required to provide more detailed data on the European flounder’s status in the Baltic Sea.

## Figures and Tables

**Figure 1 animals-13-01448-f001:**
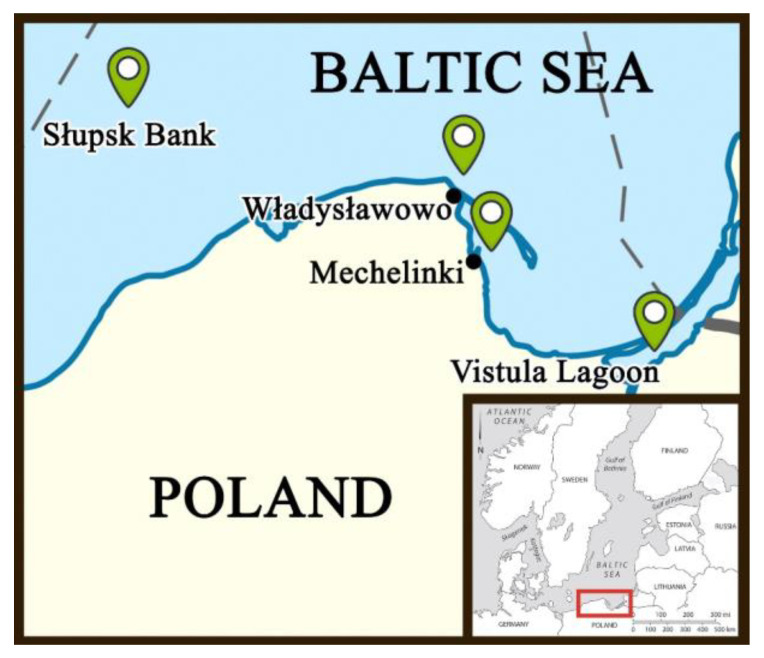
Map of the fish sampling sites along the central Polish coastline, namely, Mechelinki (54°37′07.0″ N 18°33′01.4″ E) (N = 50), Władysławowo (54°48′17.6″ N 18°24′33.1″ E) (N = 50), the Słupsk Bank (55°02′22.6″ N 16°22′03.0″ E) (N = 50), and the Vistula Lagoon (54°21′00.0″ N 19°33′00.0″ E) (N = 40).

**Figure 2 animals-13-01448-f002:**
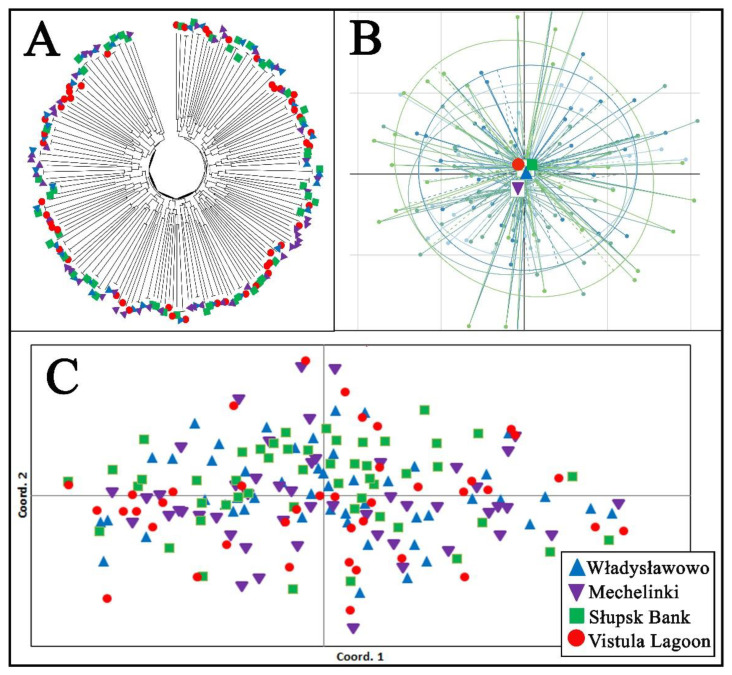
Genetic relationship among examined fish, estimated by (**A**) the unrooted neighbor-joining tree of individuals based on allele sharing distances (*DAS*), (**B**) scatter plot of the principal component analysis (*PCA*), and (**C**) principal coordinate analysis (*PCoA*), based on individual pairwise genetic distances.

**Figure 3 animals-13-01448-f003:**
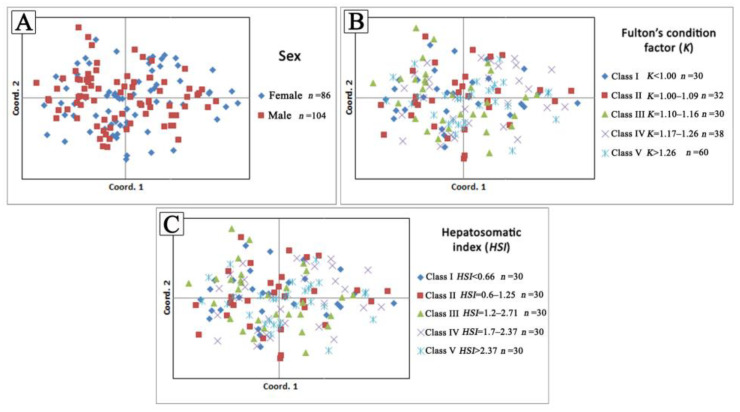
Principal coordinate analysis (*PCoA*) based on individual pairwise genetic distances between each predefined fish group based on (**A**) sex, (**B**) Fulton’s condition factor (*K*), and (**C**) Hepatosomatic index (*HSI*) values.

**Table 1 animals-13-01448-t001:** Multilocus genetic diversity parameters of European flounder (*Platichthys flesus*) individuals examined in the current study. *Ao*: observed number of alleles, *Ae*: number of effective alleles, *Ho*: observed heterozygosity, *He*: expected heterozygosity, *I*: Shannon’s index, *PIC*: polymorphism information content, and *Fis*: fixation index. Values of the H–WE test and values of the *Fis* indicator are significant at ^a^
*p* < 0.05 and ^b^
*p* < 0.01. Significant values of the H–WE test and values of the *Fis* indicator after Bonferroni correction are bolded.

Sampling Site	*Ao*	*Ae*	*Ho*	*He*	*P*	*I*	*PIC*	*Fis*
Władysławowo	18.8	9.6	0.815	0.847	0.121	2.697	0.887	0.031
Mechelinki	17.9	9.6	0.767	0.841	0.045 ^a^	2.656	0.835	0.074 ^a^
Słupsk Bank	19.4	10.8	0.751	0.844	**0.005 ^b^**	2.768	0.903	**0.093 ^b^**
Vistula Lagoon	18.8	10.1	0.801	0.844	0.214	2.579	0.832	0.036

**Table 2 animals-13-01448-t002:** Results of the Bottleneck analysis. Comparison of expected heterozygosity (*He*) vs. heterozygosity (*Heq*) expected under the infinite alleles model (*IAM*). Stepwise mutation model (*SMM*) and two-phase model of mutation (*TPM*) in examined European flounder (*Platichthys flesus*) from the Baltic Sea. *M*: Garza–Williamson Index and significant values of the Wilcoxon test for *He* > *Heq* are bolded (*p* < 0.05).

Fish Group	*He*	*IAM*		*TPM*		*SMM*		*M*
		*Heq*	*P*	*Heq*	*P*	*Heq*	*P*	
Władysławowo	0.847	0.820	**0.049**	0.846	0.578	0.858	0.986	0.600
Mechelinki	0.767	0.813	0.039	0.840	0.638	0.855	0.996	0.574
Słupsk Bank	0.844	0.812	**0.001**	0.836	0.406	0.850	0.968	0.647
Vistula Lagoon	0.653	0.817	0.044	0.843	0.658	0.856	0.991	0.587

## Data Availability

Data presented in the manuscript are available from the corresponding author upon request.

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
