# Peer review of "Population Genetic Study on the European Flounder (Platichthys flesus) from the Southern Baltic Sea Using SNPs and Microsatellite Markers"

_animals, 2023, doi:10.3390/ani13091448_

Round 1

Reviewer 1 Report

Title: Population Genetic Study on the European Flounder (Platichthys Flesus) From the Southern Baltic Sea by Application of Microsatellite DNA and SNP Markers Analysis 

The authors analyzed 19 microsatellite markers and 2 SNPs from 190 European flounders to test the level of genetic diversity and population genetic structure in the southern Baltic Sea. They further assessed the possibility of the occurrence of Baltic flounder in the study area. Overall, the manuscript is interesting and well-written. However, there are some improvements needed in the write-up and data analysis and presentation. Please find the specific comments below. 

Please write the species name according to the ICZN.

Line 30: What do you mean by 'body condition'? Is it size, mass, or what?

Line 34: How did you test whether the alteration was adverse or selective?

Introduction: There are much basic information on the biology and economic uses (for example: line 69-70: The species is commonly served by bars and restaurants along the Baltic coast, mainly in a fried form but they can also be boiled, steamed or baked) of the species that are irrelevant to the research objectives. Therefore, the introduction section needs to be more focused. 

Lines 125-126: ................fish sampled from Vistula Lagoon were exclusively young, immature at an age of 1+ and 2+ years old ...... Why were the immature ones sampled? Were they also used in the morphological analysis? How were the morphological analyses of immature and mature individuals sampled?

Lines 162-163: Were those annealing temperatures the same for all five (I-V) multiplex PCRs? 

Line 205: ..... a threshold value at P<0.1............ Please add justification, why 0.1? 

Line 242: Which genetic characteristics were used for the correlation analysis? How was it quantified? Needs to be explained in the methods clearly. Actually, in the results, I could not see any exact correlation test outcomes. Which matrices were used for the PCoA? Authors are required to check it clearly and explain what they wanted to assess and how did they execute it. This area has a major weakness in the manuscript. 

Line 278: Table 2. I think it is better to show average values here in the table for each sampling locality and move the data for the individual marker to the supplementary file. 

Line 299: How was the NJ tree constructed? Panels A and B have too low resolutions, difficult to read. 

Line 319: Table 3. Same issue as in Table 2.

Lines 324-325: ..The performed statistical test did not detect any significant correlation between biological traits and the genetic characteristics in the examined fish (Figure 3)............... Do these tests examine the correlation between the morphological and genetic variables? 

Lines 437-439: Please update the concluding statement according to the results using the proper term for the analysis you have done. 

All the best!

Author Response

Dear Reviewer,

We are grateful for the careful consideration and reviews of our original submission and the valuable feedback provided that has greatly improved the revised version of our manuscript. According to these comments the authors have made substantial changes in the revised manuscript. The detailed list of changes and responses are listed in the attached pdf file. For clarity each response was typed in red and all the changes in the revised manuscript were marked in yellow.

Best regards,

Author Response

(The authors gave the same response as above.)

Reviewer 3 Report

The current manuscript describes the population structure of European flounder in the southern coast of the Baltic sea. Four samples of 40-50 fish were collected in four sites, three in 2018 and one in 2020. The fish were genotyped at 19 microsatellites and two diagnostic SNPs. A wide range of analytical tools were applied to the data. 

Flounder were found to be a single population and the diagnostic SNPs indicated that no Baltic flounder were included in the analysis.

I appreciate this study, which albeit limited to a small region and in time, is informative of the status of European flounder in the southern Baltic. The results should be appreciated by other researchers working on European fish genetics and for managers of flounder in the southern Baltic. The manuscript is clearly written and well structured, with minor faults. The sample sizes and the numbers of markers are adequate to answer most of the questions, and the analyses are mostly appropriate. I have only two reservations regarding the analyses: since all samples belong to the same population, I find the estimation of migrants between samples odd. I don’t find that analysis is appropriate, and I'm not sure it's interpretable. Also, the analysis of Ne at sample level is also odd. Since all samples belong to the same population, I would calculate a single Ne for flounder in the southern Baltic sea. Which would be more informative for other studies on flounder.

Minor comments:

Line 331 -333 -  Avoid parenthesis, write actual sentences with subclauses.

Line 358 - Avoid ambiguous sentences such as "This study". It is often unclear if you make reference to your study or the last reference you cited. Substitute by "Here, " for example. Check all occurrences of "This study".

Line 400-401 - I think this statement is farfetched. You have a single sample from 2020, I don’t think you have enough data to pronounce yourself on Ne changes over the two year period. Which you acknowledge. I would remove such statement.

Line 418 - there ARE null alleles

Author Response

(The authors gave the same response as above.)

Round 2

Reviewer 1 Report

Thank you for improving the manuscript and incorporating all the suggestions.  

ALl the best!

Author Response

Dear Editor,

Thank you again for a lot of valuable suggestions and we greatly appreciate your positive assessment of our study!

Best regards,

Reviewer 2 Report

While I recognize that manuscript has been improved, authors are still using not appropriate terms, such as, for example, "genetic characteristics" - this is a population genetic  study - patterns of genetic diversity and population structure. 

The authors should additionally discuss and take into consideration neutral possibilities, as I said in my first review - In severely overexploited populations, where fixation of harmful alleles can lower population viability and increase the danger of extinction, genetic variety loss owing to drift is particularly relevant. Please add a line or paragraph to the discussion and introduction sections.

Author Response

Dear Reviewer,

We are grateful for the careful consideration and reviews of our original submission and the valuable feedback provided that has greatly improved the revised version of our manuscript. According to these comments the authors have made substantial changes in the revised manuscript. The detailed list of changes and responses are listed in the attached pdf file. For clarity each response was typed in red and all the changes in the revised manuscript were marked in green.

Best regards,
